PhyloSift: phylogenetic analysis of genomes and metagenomes

Darling Aaron E. 1 2 aarondarling@ucdavis.edu
Jospin Guillaume 2
Lowe Eric 2
Matsen Frederick A. IV 5
Bik Holly M. 2
Eisen Jonathan A. 3 4
1 ithree institute, University of Technology Sydney , Sydney , Australia
2 Genome Center, University of California , Davis, CA , United States of America
3 Department of Evolution and Ecology, University of California , Davis, CA , United States of America
4 Department of Medical Microbiology and Immunology, University of California , Davis, CA , United States of America
5 Fred Hutchinson Cancer Research Center , Seattle, WA , United States of America
Moustafa Ahmed
Electronic publication date: 2014 Jan 9
Publication date: 2014
Volume: 2
Electronic Location ID: e243
Received 2013 Mar 21; Accepted 2013 Dec 19
Copyright: © 2014 Darling et al.
Copyright year: 2014
Copyright holder: Darling et al.
License: This is an open access article distributed under the terms of the Creative Commons Attribution License, which permits unrestricted use, distribution, and reproduction in any medium, provided the original author and source are credited.
License URL: https://creativecommons.org/licenses/by/3.0/

Keywords: Metagenomics, Phylogenetics, Forensics, Bayes factor, Microbial diversity, Community structure, Microbial ecology, Edge PCA, Phylogenetic diversity, Microbial evolution

Funding: US Department of Homeland Security contract HSHQDC-11-C-00091 This work was performed under contract HSHQDC-11-C-00091 from the US Department of Homeland Security. The funder had no role in study design, data collection and analysis, decision to publish, or preparation of the manuscript.

==============================
Like all organisms on the planet, environmental microbes are subject to the forces of molecular evolution. Metagenomic sequencing provides a means to access the DNA sequence of uncultured microbes. By combining DNA sequencing of microbial communities with evolutionary modeling and phylogenetic analysis we might obtain new insights into microbiology and also provide a basis for practical tools such as forensic pathogen detection.

In this work we present an approach to leverage phylogenetic analysis of metagenomic sequence data to conduct several types of analysis. First, we present a method to conduct phylogeny-driven Bayesian hypothesis tests for the presence of an organism in a sample. Second, we present a means to compare community structure across a collection of many samples and develop direct associations between the abundance of certain organisms and sample metadata. Third, we apply new tools to analyze the phylogenetic diversity of microbial communities and again demonstrate how this can be associated to sample metadata.

These analyses are implemented in an open source software pipeline called PhyloSift. As a pipeline, PhyloSift incorporates several other programs including LAST, HMMER, and pplacer to automate phylogenetic analysis of protein coding and RNA sequences in metagenomic datasets generated by modern sequencing platforms (e.g., Illumina, 454).

Introduction

Metagenomics – the sequencing of DNA isolated directly from the environment – has become a routinely used tool with wide applications (Thomas, Gilbert & Meyer, 2012). Used primarily in the study of microorganisms, metagenome sequencing has now been carried out on a variety of environments where one finds microbes — from plants and animals to every kind of natural and man-made environment around the globe. Metagenomic sequencing has provided fundamental insight into the diversity of microbes and their function and roles in ecosystems. Initially, metagenomics was used largely as a way of simply obtaining some genomic information about organisms for which culturing technique was unknown (Béjà et al., 2000). However, due to decreases in the cost and the difficulty of sequencing, metagenomics has become a tool for studying any microbial community, regardless of cultivability.

One strength of metagenomic approaches arises from the ability to sample the genomes of organisms in a particular environment approximately uniformly at random. This effect is achieved with the random “shotgun” sequencing methods originally applied for de novo genome sequencing of individual organisms (Venter et al., 2004; Tyson et al., 2004). From random shotgun sequence data of DNA isolated from environmental samples, one can make inferences about what organisms are present in a sample (i.e., who is there?) as well as their functional potential (i.e., what are they doing?). In addition, by comparing shotgun metagenomic data across samples one can study larger scale issues such as ecology and biogeography and also attempt to correlate particular organisms or functions with “metadata” about samples (e.g., health status, nutrient cycling rates, etc Tringe et al., 2005). Furthermore, by sampling a community directly one can avoid certain problems inherent in culturing such as contamination, population bottlenecking, and taxonomic bias (Eisen, 2007). In this sense metagenomics can be considered an extension of “culture-independent” ribosomal RNA gene surveys (Hugenholtz, Goebel & Pace, 1998). The great potential for novel insight into microbial communities has led researchers in fields as diverse as medicine and agriculture, law enforcement, biodefense, ecology, evolution, and industry to apply metagenomic methods.

Athough great potential exists for metagenomics to yield insight into the hidden world of microbes, many challenges remain before this potential can be realized. Perhaps the biggest challenges lie in analysis of the data (Chen & Pachter, 2005). First, metagenomic samples reflect entire communities of organisms, unlike “traditional” genome sequencing of individual organisms or clones (i.e., from cultures of a single isolate where genetic diversity has undergone a bottleneck). The large number of microbial taxa in environmental samples can be a challenge for some types of analysis. Within-species genomic polymorphism presents an even greater challenge (Kunin et al., 2008). This challenge arises largely because shotgun metagenomic sequencing protocols destroy some of the most valuable information present in a sample: genetic linkage. Loss of linkage information occurs in two ways: during sample extraction and fragmentation of DNA for sequencing. In nearly all metagenomic sample processing methods, cells from the microbial community are lysed together to obtain a common pool of DNA. This practice causes DNA from many different cells to mix together, so that the cellular compartmentalization of individual genotypes is destroyed. Subsequently, long chromosome-scale DNA fragments are typically broken by mechanical or enzymatic means into fragments small enough for processing with current sequencing protocols. The resulting sequenced fragments are usually less than 1 kbp in length. Although it is possible to generate data for larger fragments via cloning (Béjà et al., 2000) or using Pacific Biosciences sequencing, most metagenomic data is currently being generated with short read/short insert sequencing chemistry such as that offered by Illumina. Though short read methods are quicker, easier, and lower cost per base and per read than large fragment approaches, there is a tradeoff in information quality. The shearing results in further loss of genetic linkage information, since we no longer have direct information on how short DNA fragments are arranged into chromosome-scale molecules.

The lack of linkage information limits the ability to use metagenomic data for phylogenetic and population genetic analysis, since most current methods assume complete linkage information is available. In practice, improved sample processing methods could potentially retain the genetic linkage information of a microbial community throughout the sequencing process. High throughput single-cell genomics (e.g., applied to hundreds or thousands of cells) offers an alternative to the standard metagenomics workflow that preserves information about the compartmentalization of genetic material into cells (Woyke et al., 2010; Lasken, 2012; Rinke et al., 2013). However, single cell approaches are still limited in their utility by a number of technical issues including contamination, expensive and extensive equipment needs, missing data, and the creation of chimeras (Blainey, 2013); they will always be more limited in throughput than their standard metagenomics counterparts.

Thus the research community is left with developing and using computational methods to sift through and make sense of short read, random shotgun metagenomic data. Though there are many important steps in analyzing metagenomic data, we believe that a critical component is phylogenetic analysis of the sequences. Among the uses of phylogenetic analysis in metagenomics are: improved classification of sequences using phylogenetic methods, functional prediction for genes, alternative metrics of alpha and beta diversity, improved identification of operational taxonomic units (OTUs), and sequence binning (Meyer et al., 2008; Matsen, Kodner & Armbrust, 2010; Evans & Matsen, 2012; Kembel et al., 2011; Matsen & Evans, 2013; Stark et al., 2010; Wu & Scott, 2012; Brady & Salzberg, 2009; Brady & Salzberg, 2011; Sharpton et al., 2011; Jolley et al., 2012; Sunagawa et al., 2013; Eisen, 2012).

In the present manuscript, we introduce PhyloSift, a new method for phylogenetic analysis of metagenomic samples and for comparison of community structure among multiple related samples. The new method leverages phylogenetic models of molecular evolution to provide high resolution detection of organisms in a metagenome. Our approach is based on well known statistical phylogenetic models, is amenable to Bayesian hypothesis testing, and uses name-independent and OTU-free analyses to provide higher resolution about microbial community assemblages (versus methods that rely on taxonomy or OTUs). These methods can be applied to any single phylogeny at a time, and expand on our previous experience building AMPHORA (Wu & Eisen, 2008). We additionally propose a set of 37 “elite” marker gene families that have largely congruent phylogenetic histories, thus improving the limit of detection for rare organisms in microbial communities. We contribute an open-source implementation of the method that has been engineered for ease-of-use on 64-bit Linux and Mac platforms. Finally, we compare the features of PhyloSift to some related methods to provide readers with insight into when use of our approach is and is not appropriate.

Previous work

Estimating community composition from amplicon data

High throughput sequencing of marker gene amplicons (homologous loci such as 16S/18S rRNA) has emerged as a powerful and straightforward means to analyze microbial community structure. In contrast to shotgun metagenomics, amplicon approaches currently make the detection of rare taxa easier and require less starting genomic material than some metagenomic approaches, although transposon-catalyzed libraries have been generated from as little as 30 pg total material (Adey et al., 2010). By design rRNA surveys offer a “standardized” snapshot of microbial communities with reads from a single or small number of genes, considerably simplifying the tasks of alignment and analysis. Amplicon studies generally focus on characterizing and comparing microbial community structure without much analysis of functional gene repertoire.

A variety of software pipelines can be used to process and analyze rRNA amplicon data (Bik et al., 2012). Inferring microbial assemblages typically relies on clustering of Operational Taxonomic Units (e.g., at a 97% sequence identity cutoff, using either de novo or reference-based clustering), where taxonomy is assigned to representative sequences using either BLAST searches or the RDP classifier (a Naive Bayesian Classifier Wang et al., 2007). Users can subsequently carry out a suite of downstream ecological and diversity analyses, including rarefaction (e.g., analyses for Chao1 estimation, OTU richness, or phylogenetic diversity as implemented in QIIME Caporaso et al., 2010), and Principal Component Analysis and Jackknife cluster analysis (e.g., using phylogeny-derived UniFrac distances Lozupone & Knight, 2005).

Amplicon approaches are now relatively cheap and easy to carry out. However some computational bottlenecks hinder fine-scale analysis of amplicon data. Analysis pipelines cannot readily distinguish rare members of a microbial community from noise in data caused by sequencing errors or chimeric reads (Bik et al., 2012). The RDP classifier (Wang et al., 2007) provides a statistical method for assessing confidence in taxonomic classifications. Any of these methods are limited relative to phylogenetic methods, in that they can only distinguish named groups of organisms and are limited to the resolution of the taxonomy.

Community composition from metagenomes

Methods have also been developed to estimate and analyze community composition from metagenomic data sets. These methods typically focus on a small subset of widely conserved marker genes mined from metagenomic sequence reads, usually representing 1% of any given shotgun dataset. Marker genes include well-characterized protein coding genes (e.g., ribosomal proteins or elongation factor genes) or conserved noncoding regions (e.g., rRNA). A variety of computational approaches are now available to investigate the community composition of metagenome datasets, including: AMPHORA (bacterial protein markers and tree insertion via parsimony) (Wu & Eisen, 2008) and AMPHORA2 (bacterial/archael protein and DNA markers and tree insertion via likelihood or parsimony) (Wu & Scott, 2012), MLTreeMap (reference gene families with taxonomic and functional information and tree insertion via maximum likelihood) (Stark et al., 2010), MetaPhyler (taxonomic classifiers for each of the reference marker genes published in the AMPHORA set) (Liu et al., 2010), EMIRGE (an expectation-maximization method to reconstruct rRNA genes from metagenome data and estimate taxon abundance) (Miller et al., 2011), and PhylOTU (phylogenetic methods to mine rRNA and define OTUs from metagenome data) (Sharpton et al., 2011).

An interesting alternative approach is employed by the software MetaPHlan (Segata et al., 2012), which instead of using universally conserved genes, employs a database of clade-specific genes to estimate abundance of known taxonomic groups. This approach may work well in environments where the genomic diversity is very well characterized.

Community composition analysis from metagenomes has some potential advantages over amplicon studies. For example, metagenome sequencing might avoid bias introduced by preferential binding of PCR primers to DNA from some organisms in amplicon studies and can also capture genomes from organisms which lack amplicon target genes, such as viruses. Whole-metagenome surveys also have the potential to provide insight into enzymatic and other functional processes in microbial communities, and so a single dataset can provide both community composition and functional information. One major limiting factor is that reference genome databases have narrow phylogenetic breadth relative to marker genes (e.g., rRNA) (Wu et al., 2009).

Taxonomic classification of metagenome sequences

Current methods for taxonomic classification of metagenomic sequences generally leverage one or two information sources: sequence composition and/or sequence identity to reference databases. Some existing composition classifiers include TACOA (supervised classification using k-nearest neighbors) (Diaz et al., 2009), PhyloPythia (McHardy et al., 2006) and PhyloPythiaS (multiclass support vector machine classifier using oligonucleotide frequencies) (Patil et al., 2011), NBC (Naive Bayesian Classifier) (Rosen, Reichenberger & Rosenfeld, 2011), and Eu-Detect (oligonucleotide binning to separate eukaryote sequences in feature vector space) (Mohammed et al., 2011), although this is not an exhaustive list. Related methods such as Self-Organizing Maps (e.g., eSOMS (Dick et al., 2009)) can be applied to tetranucleotide frequencies in combination with other information sources such as contig coverage/abundance information to produce visual “maps” displaying different bins, although this does not result in taxonomic assignment.

Identity-based classification methods compare metagenome sequences against reference databases to identify putative homologs. Examples of current identity-based classification tools include MEGAN (a Lowest Common Ancestor algorithm that summarizes BLAST outputs to assign taxonomy) (Huson et al., 2007), SORT-Items (reciprocal BLAST approach to detect significant orthology) (Haque et al., 2009), MTR (a variation on Lowest Common Ancestor approaches that uses multiple taxonomic ranks) (Gori et al., 2011), and ProViDE (analysis of alignment parameter thresholds, specifically customized for classifying viral sequences) (Ghosh et al., 2011). Some approaches are able to combine both sequence identity and composition when classifying (Brady & Salzberg, 2009; Brady & Salzberg, 2011). Again, this is not an exhaustive list.

As the focus of our current work is on phylogenetic analysis rather than taxonomic classification, we do not discuss the relative merits of each approach to taxonomic classification in detail, nor do we provide benchmarks of taxonomic classification methods.

Methods

PhyloSift implements a method for analyzing microbial community structure directly from metagenome sequence data. Figure 1 gives an overview of the analysis workflow as executed when analyzing a metagenomic sample. The analysis can be decomposed into four stages: 1. searching input sequences for identity to a database of known reference gene families; 2. adding input sequences to a multiple alignment with reference genes; 3. placement of input sequences onto a phylogeny of reference genes; and 4. generation of taxonomic summaries. We now describe the details of each step along with our design decisions and rationale.

Figure 1 PhyloSift client workflow.

This workflow is applied to the user’s sequence data. DNA input sequences are processed via both the rRNA and protein parts of the workflow.

Reference gene families used by PhyloSift

The standard PhyloSift database includes a set of 37 “elite” gene families previously identified as nearly universal and present in single-copy. These 37 gene families are a subset of the 40 previously reported (Wu, Jospin & Eisen, 2013), with three families excluded because they frequently have partial length homologs in some lineages. These “elite” families represent about 1% of an average bacterial genome, as estimated from current genome databases. In other work we have demonstrated that phylogenetic trees reconstructed on individual genes in this set are generally congruent with each other (Lang, Darling & Eisen, 2013; Rinke et al., 2013), suggesting that concatenating alignments of these families will yield a valid and more powerful estimate of their phylogenetic history. Other groups have also demonstrated that trees inferred from concatenate alignments demonstrate the least conflict with trees inferred separately from other microbial amino acid sequences (Abby et al., 2012). During the database update process (described below), these gene families are automatically extended to include putative homologs from eukarya and some viruses with large genomes such as the Mimivirus. Most small viral genomes lack homologs of these gene families.

In addition to the elite 37 families, the PhyloSift database also includes four additional sets of gene families:

• 16S and 18S ribosomal RNA genes

• mitochondrial gene families

• Eukaryote-specific gene families

• Viral gene families

Combined, this yields a set of approximately 800 gene families in the standard PhyloSift database, most of which are viral.

Detailed PhyloSift client workflow

Sequence identity search

This first step in a PhyloSift analysis aims to identify regions of the input sequences that may be homologous to gene families in the reference database. Input sequences to this step can be of any length ranging from short 30nt next-generation sequence reads to fully assembled genomes or metagenomes. Recognized input formats include FastA and FastQ (paired, unpaired, phred33, phred64, and/or interleaved pairing), and these can optionally be supplied as bzip2 or gzip compressed data files. Sequence input can be streamed via stdin or unix named pipes. Amino acid input sequences can also be processed.

PhyloSift uses LAST (Kiełbasa et al., 2011) for sequence similarity search against the reference databases. We evaluated many possible search algorithms and implementations before finally selecting LAST. Other options we evaluated were BLAST (Altschul et al., 1997) v2.2.23, BLAST+ (Camacho et al., 2009) v2.2.28+, and RAPsearch2 (Zhao, Tang & Ye, 2011) v2.04, and bowtie2 (Langmead et al., 2009) v2.0.0-beta5. Given the large volume of sequence data that must be processed, a key evaluation criterion was algorithm efficiency both in CPU time and memory requirements. A second criterion is the ability to perform six-frame translated searches of DNA sequence against an amino acid database with the possibility to tolerate frame-shift errors in the sequence. Among the evaluated methods, BLAST and BLAST+ were slowest (data not shown) and frameshift detection was non-functional in the version of BLAST+ we obtained from NCBI. We excluded these from further consideration. RAPsearch2 was much more computationally efficient than either BLAST or BLAST+, but the version we obtained could not process sequences > 1 kbp and did not support frameshift detection. In our testing, LAST was able to process sequence data as quickly as RAPsearch2 (e.g., orders of magnitude more quickly than BLAST) and supports both frameshift detection and input sequences of arbitrary length. LAST also supports all three of the primary search types we require: DNA vs. DNA, DNA vs. AA, and AA vs. AA. We also evaluated bowtie2, a program typically used for mapping reads to a reference genome, for the purpose of screening reads against a database of noncoding RNA sequences (currently 16S and 18S). bowtie2 does not offer translated amino-acid searches. Relative to LAST, bowtie2 is able to identify similarity to the RNA database sequences more quickly. However, even though the speedup over LAST was substantial (data not shown), the compute time saved is small relative to the total time consumed in the complete PhyloSift client workflow. Therefore we decided to use only LAST since using only a single local alignment search tool simplifies the code. One shortcoming of LAST is that current versions do not support multithreaded parallelism. PhyloSift implements optional process-level parallelism by spawning multiple LAST searches against the protein database.

One feature of reference gene family sequences being searched at this stage bears special mention. During database construction (described elsewhere) a representative subset of all available sequences are selected from each gene family to be searched in the search stage. These representatives are chosen to span the phylogenetic diversity of the gene family without including closely related sequences (see Section “PhyloSift database update workflow”). This is important because it reduces the volume of sequence to search and because part of LAST’s fast heuristic to identify candidate regions to align involves eliminating redundant and repetitive k-mers from the search space (Kiełbasa et al., 2011). Thus, a database constructed with all sequences (and not just divergent representatives) could in principle reduce sensitivity in aligning reads to those database sequences.

The search stage identifies a set of candidate amino acid sequences from the input data that are similar to reference gene families. If DNA was provided as input the corresponding DNA sequences are also reported.

Alignment to reference multiple alignment

Prior to the alignment stage all input sequence regions with putative homology to reference gene families have been identified and extracted. In this stage, each candidate sequence is added to an amino acid or RNA multiple sequence alignment of the reference gene family. If the input sequences were DNA, a codon multiple sequence alignment congruent to the amino acid alignment is also generated.

PhyloSift applies the hmmalign program from the HMMER 3.0 software package (Eddy, 2011) to add the candidate sequences to reference multiple sequence alignments. During construction of the PhyloSift reference database (described in section “Custom gene families”) a profile-HMM is generated from a multiple alignment of the gene family reference sequences. When processing candidate sequences, PhyloSift then uses the profile-HMM to map the input sequence to the reference multiple alignment. Application of a profile-HMM to align highly divergent sequences suffers some documented shortcomings, in particular that alignment accuracy decreases with divergence of source sequences used to construct the profile-HMM (Löytynoja, Vilella & Goldman, 2012). This is one avenue for future improvement of PhyloSift and protein evolution models in general.

Finally, PhyloSift concatenates the alignments of the 37 elite markers to a single multiple sequence alignment. When a single input sequence aligns to multiple genes, the aligned sequence becomes a single row in the concatenated alignment. All other sequences are represented in separate alignment rows.

PhyloSift treats input sequences with similarity to non-coding RNAs differently than protein genes. Sequences longer than 600nt are aligned using Infernal’s cmalign program with the global alignment option. Short sequences are aligned with hmmalign to a profile-HMM of the non-coding RNA molecule. Although the profile-HMM does not capture secondary structure, the alignment computation is significantly faster with currently available versions of Infernal and HMMER. In our experience a banding threshold (a parameter that determines the size of the search space and hence amount of computational effort) of 1 × 10−20 is required to obtain accurate local alignments with Infernal for short sequences, but this requires several minutes of CPU time per aligned sequence, which is not practical when aligning millions of amplicon sequences.

Placement on a phylogenetic reference tree

At this stage, aligned input sequences are placed onto a phylogenetic tree of the reference sequences. PhyloSift employs pplacer (Matsen, Kodner & Armbrust, 2010) for this task. pplacer can be run in either maximum likelihood (ML, the default) or Bayesian mode. When run in ML mode, pplacer identifies and reports a set of most likely attachment points for each aligned sequence to the reference phylogeny, as well as a “likelihood weight ratio” representing the relative likelihood for the chosen attachment point over other possible attachment points.

When run in Bayesian mode, pplacer calculates the posterior probability that the query sequence diverged from particular branches of the reference tree via direct integration. In contrast to ML placement which selects a single most likely attachment point, the branch posterior probability integrates over all possible attachment points for the query sequence on the branch. The posterior probability is used when calculating Bayes factors for lineage tests, described below.

Taxonomic summary of read placements

At this final stage of analysis, PhyloSift summarizes the phylogenetic placements in a human-friendly format. For each gene family, the PhyloSift database contains a gene-tree/taxonomy reconciliation encoding a pre-computed mapping of edges in the gene family phylogeny to edges in the NCBI taxonomy. The method used to calculate these reconciliations is described in the database update workflow section, below.

Input to this stage of analysis is one or more “jplace” format (Matsen et al., 2012) files containing an edge-labeled reference tree for a gene family along with a collection of one or more sequence placements onto that tree. Information about each sequence’s placement consists of the log-likelihood of placement at several (usually up to 7, a configurable limit) of the highest likelihood edges on the reference tree, along with the probability mass that the sequence belongs at that position of the tree, and finally the weight of the sequence. When analyzing unassembled reads the sequence weights are typically always 1, when analyzing assembled contigs the weights may be set to a value based on estimated depth-of-coverage for that contig.

PhyloSift parses each of the jplace files and uses the gene-tree/taxonomy reconciliation to convert probability mass over read placements into a probability mass over the taxonomy, summing these masses over all reads and gene families. Any particular edge in the gene tree may be mapped to many equally optimal locations in the taxonomy. PhyloSift distributes the placed sequence’s mass equally among all optimal locations.

Finally, PhyloSift reports the summarized taxonomy probability mass distribution in a variety of formats.

Visual presentation of taxonomic summary

For easy visualization and exploratory data analysis, PhyloSift produces Krona plots (Ondov, Bergman & Phillippy, 2011) showing taxonomic probability mass in the 37 elite gene families, and a separate Krona plot showing taxonomic probability mass distribution summed across the elite families and all other families.

Figure 5 provides an example of PhyloSift’s Krona reports.

Parallelism and stream computing

PhyloSift supports streaming input of sequences, this permits analysis to proceed as sequences arrive over a network connection, for example.

Comparison among samples

One of the unique aspects of PhyloSift relative to other methods for comparative metagenomics is that the phylogenetic approach we have implemented enables direct comparison of the phylogenetic structure and relative abundance of metagenome samples without resorting to taxonomic relative abundance estimates. Perhaps the most powerful exploratory data analysis tool for comparing community structures among samples is Edge Principal Component Analysis, or edge PCA (Matsen & Evans, 2013). Edge PCA applies the standard dimensionality-reduction tool of PCA to a matrix where columns correspond to edges in the reference phylogeny, rows correspond to each sample, and each entry is the difference in placed sequence probability masses on either side of that edge. When applied in this manner, the eigenvalues of each eigenvector that results from PCA correspond to weights indicating how important each edge in the reference phylogeny is for explaining the variation among samples in that dimension. These eigenvectors can be naturally visualized as thickened branches along the reference phylogeny (Matsen & Evans, 2013).

PhyloSift includes the guppy program from pplacer, which in addition to edge PCA also provides means for hierarchical clustering of multiple samples using an algorithm specialized to the case of masses on a tree, calculation of Kantorovich-Rubenstein distances among samples (Evans & Matsen, 2012), and other tools for calculating sample summary statistics such as weighted phylogenetic diversities.

PhyloSift database update workflow

An integral component of PhyloSift is an automated means to update the gene family database with newly sequenced genomes. Genome databases continue to grow quickly, with, on average, dozens of new genome sequences becoming available every week. The quality of these genomes can be highly variable, ranging from low-quality drafts to nearly finished sequence. PhyloSift’s database update mechanism incorporates some basic quality control measures.

Acquiring new genome data

The PhyloSift database update module maintains a local repository of all known and processed genomes. Upon initiating a new update, the database update module identifies any new genomes available in the NCBI finished, NCBI draft, NCBI WGS, and EBI viral, organelle, bacterial, archaeal, and eukaryal databases. Any new genomes are fetched and stored in the local repository.

Gene family search and alignment workflow on each genome

In this stage, the search and alignment stages of the previously described PhyloSift client workflow are run for each new genome. After this stage, the regions from each new genome that are highly similar to gene families in the database are identified, extracted, and aligned using the family’s profile-HMM. A complete multiple alignment for each family is then created by adding the aligned regions from each genome to a single multiple alignment file. Because each region has been aligned to the same profile-HMM (or covarion model for noncoding genes) and non-aligning sites in the query genome removed, generation of a new multiple alignment is a simple matter of concatenating the individual alignments.

PhyloSift also generates codon alignments for each protein-coding gene family at this stage by replacing amino acids with their codons and replacing each gap with a gap triplet.

We note that profile-HMMs are not recomputed during the database update, thereby avoiding problems with model drift.

The PhyloSift reference database is available independently of the software at the following location: http://edhar.genomecenter.ucdavis.edu/~koadman/phylosift_markers.

Phylogenetic inference and pruning

The next step of database update involves constructing a phylogenetic tree for each gene family. Currently PhyloSift employs FastTree 2.1 (Price, Dehal & Arkin, 2010) to generate approximate maximum likelihood trees for this task. PhyloSift also infers separate trees for the codon and amino acid alignments of each gene family.

Reference databases frequently contain genomes for a multitude of closely related strains. In many gene families, the gene sequences present in genomes of closely related strains may be identical to each another. Identical gene sequences would create uncertainty in the placement of reads in a strain group. In order to reduce compute time and memory requirements, closely related sequences are pruned from the PhyloSift reference database. Pruning is done with an algorithm that maximizes phylogenetic diversity of the sequence set without including any sequence pairs separated by fewer than X amino acid (or nucleotide) substitutions per site, where X is a configurable variable with default value 0.01.

Selection of representatives for similarity search

The PhyloSift client workflow uses LAST to search for similarity between input sequences and reference sequences. During the database update the set of reference sequences is updated to include representatives of any newly sequenced genomes. As above, we select a subset of sequences that maximize phylogenetic diversity while requiring sequence pairs to be separated by at least X amino acid substitutions per site. In this case, X defaults to 0.1.

Taxonomic reconciliation

Many of the data sources for new genomes provide a taxonomic identifier for the genome that places it in the NCBI taxonomy. Throughout the database update process, the associations between taxon ID and individual sequences are maintained. The tips of reconstructed phylogenies can therefore have some or all nodes annotated with the taxon ID associated with that tip. Given this information, PhyloSift generates a mapping of edges (e.g., the edge above each node) in the gene tree phylogeny to edges in the taxonomic tree. To do so, we first compute the split (bipartition) encoding of the gene tree and the taxonomic tree. A tree’s split encoding is simply the set of splits encoded by each edge in the tree, where the split for edge i is a binary vector Si = {si,1...si,n}, si,j∈{0, 1}. Here n is the number of leaf nodes shared by the two trees. For convenience, we denote the split encoding for the gene tree as S(G) and use S(T) for the taxonomic tree. Then for each edge i in the gene tree, we compute its mapping Mi to taxonomic tree edges as: Mi=argminSj∈STHSiG,Sj

where H(⋅, ⋅) is defined as the Hamming distance among equal-length binary vectors. We note that there may be many possible edges in S(T) with equally minimal Hamming distance to an edge i in S(G). In this case Mi includes all of these edges, and so Mi⊆S(T) and |Mi| ≥ 1. In the client workflow when assigning placement probability mass to names, the placement mass on edge SiG is divided equally among the taxonomic groups associated with Mi. Finally, we discard highly ambiguous mappings where |Mi| > y. Here y is an ad-hoc threshold with a default value of 30. These gene tree edges are labeled “Unclassifiable” due to their extreme topological discordance with the NCBI taxonomy.

Custom gene families

PhyloSift also supports the addition of custom gene families to its database. To add a gene family to the database, a multiple sequence alignment must be provided. Optionally, a table mapping each sequence identifier in the alignment and its NCBI taxon ID may also be provided. Given these inputs, PhyloSift will construct a phylogenetic tree, create a pruned set of representative sequences for similarity searching, construct a profile-HMM for alignment, and if taxon information was provided will also compute a reconciliation between the gene tree and taxonomy. The tree-building and reconciliation steps follow the approach outlined above in the PhyloSift database update workflow, with the exception that codon alignments are not generated. The resulting data is called a “package,” and is copied into the user’s PhyloSift database. The new package will be automatically included in any future runs of the PhyloSift client workflow.

Results

Bayesian hypothesis testing for the presence of phylogenetic lineages

For various applications (e.g., microbial forensics) a practitioner might want to test for the presence of a particular lineage of interest in a metagenomic sample. Phylogenetic analysis of metagenomic reads has the potential to offer resolution beyond what would be available from taxonomic methods for metagenomics. Whereas taxonomic methods can provide resolution at specific levels in the taxonomic hierarchy, such as species, genus, etc., phylogenetic methods might be able to distinguish different subtypes of named species or novel lineages at higher taxonomic levels. Phylogenetic methods are limited only by the resolution of the reference genome phylogeny and not by the resolution of manually curated taxonomies. Phylogenetic inference has the further advantage that it is based on a statistical model of sequence change where the marginal likelihood of the data given the model P(D|M) is well defined, making it possible to conduct model-based hypothesis tests using phylogenies. Taxonomic analysis methods for metagenomics are frequently based on machine learning classification methods which do not always lend themselves to such hypothesis testing.

PhyloSift provides a means to conduct Bayesian hypothesis testing for the presence of one or more query sequences belonging to organisms that have diverged along specific branches of the reference phylogeny. In order to describe the Bayesian hypothesis test we introduce the following notation: assume we are given a reference phylogenetic tree T consisting of n > 1 branches {t1…tn}. Further assume we are given a collection S of sequences s1…sm which are homologous to and aligned to the sequences at the leaf nodes of the reference phylogeny. We denote the marginal likelihood that a particular sequence sj diverged along branch ti of the reference phylogeny as P(sj∣ti). Calculation of this marginal likelihood is implemented in the pplacer software and described elsewhere (Matsen, Kodner & Armbrust, 2010).

The null hypothesis we wish to test is that there are no sequences diverging from a set of one or more lineages of interest Tx⊆T. We can express the marginal likelihood of the null hypothesis M0 as: (1) PD|M0=∏sj∈S1−∑ti∈TxPsj|ti

which can be interpreted as the product over all sequences of the probability that the sequence does not derive from a lineage of interest in Tx. The marginal likelihood of the alternative hypothesis, e.g., that one or more reads derive from a lineage in Tx, can simply be expressed as: (2) PD|M1=1−PD|M0

Using these marginal likelihoods we can construct a Bayes factor: (3) K=PD|M0PD|M1

The Bayes factor K can then be interpreted with respect to how strongly the null hypothesis is rejected by the data.

The current version of PhyloSift supports application of Bayesian hypothesis tests to a concatenated alignment of the 37 elite gene families or any other single marker gene, and can be applied to phylogenies inferred either from amino acid or codon-aligned DNA sequences.

Community structure comparison: application to human microbiome data

In addition to hypothesis testing for lineages, PhyloSift also provides a platform to conduct comparative analysis of microbial community structure directly from metagenomic data. To understand how community structure analysis with PhyloSift compares to similar analysis based on 16S rRNA amplicon sequencing we study a recently published human microbiome dataset where samples were sequenced both by a 16S amplicon and a shotgun metagenome approach (Yatsunenko et al., 2012). In that study, fecal material was collected from infants and adults at diverse geographical locations and subjected to sequencing. Over 600 samples were sequenced using the 16S amplicon protocol. Of those 106 were also subjected to metagenomic shotgun sequencing using 454 pyrosequencing chemistry. Here we apply PhyloSift to the 106 metagenomic samples and conduct a community structure comparison among the samples, and replicate the Yatsunenko et al. QIIME analyses on this subset of data.

All QIIME analyses were carried out using release 1.5.0 of the QIIME software toolkit, using the workflow and parameters reported by Yatsunenko et al. The Greengenes reference database (collapsed at 97% identity) was used to carry out a closed-reference OTU picking protocol at 97% sequence identity with uclust. All reads which matched database sequences at this level were retained for downstream processing, while non-matching sequences were excluded from further analyses. Parameters for the pick_otus.py script were as follows: –max_accepts 1 –max_rejects 8 –stepwords 8 –word_length 8. Taxonomic assignments for OTUs were given by the Greengenes database. Rarefaction and PCoA analyses were carried out using the alpha_diversity.py and beta_diversity_through_plots.py workflows. A full list of these QIIME commands and output files have been publicly deposited in figshare (http://dx.doi.org/10.6084/m9.figshare.650869).

PhyloSift processed each of the 106 samples, requiring an average of 2.5 h per sample on a single 2.27 GHz Intel Xeon E5520 core (circa 2009 model). The majority of CPU time is spent in phylogenetic placement of reads. These samples have 154,485 non-human sequence reads on average, for an average of 52 Mbp of sequence data per sample.

Figure 2 Comparison of QIIME PCA and edge PCA analysis of human fecal samples.

Samples from 106 individuals were analyzed by PCA to evaluate trends in community composition with respect to host age. 16S rDNA amplicon data and metagenomic data from the same samples was processed using QIIME and PhyloSift. QIIME analyzed the amplicon data (top left) and 16S rDNA reads extracted from the metagenomic data (top right) using a reference-based OTU picking strategy. PhyloSift analyzed the same metagenomic 16S rDNA reads (bottom left) and reads matching the 37 elite gene families (bottom right). Each PCA approach gives qualitatively similar results, differences as quantified by Procrustes analysis are given in Table 1.

We then conducted Edge Principal Components Analysis (PCA) using the reads placed onto the phylogeny of elite gene families. Edge PCA identifies the combination of phylogenetic lineages that explain the greatest extent of variation in the microbial communities in each sample. The resulting PCA plot is shown in Fig. 2, with each sample colored according to the age of the human host at the time of sampling. The PCA reveals a strong association between age and microbial community structure. This relationship was also identified by Yastunenko et al. using 16S rRNA analysis on a set of >600 samples which included the 106 studied here. In order to quantify the degree of similarity between the PhyloSift Edge PCA and QIIME PCoA results, we calculated Procrustes distances among each pair of analyses, the results are given in Table 1. In general we find that QIIME’s PCoA analysis of metagenomic 16S reads produces results that are very different to all other methods, whereas results produced by QIIME PCoA analysis of 16S amplicon data are more similar to results produced by PhyloSift on metagenomic data.

Figure 3 Lineages contributing variation in human fecal sample community structure.

106 metagenomic samples were processed using PhyloSift and their community composition compared using Edge PCA (Matsen & Evans, 2013). Lineages that decrease in abundance along the principal component axis are shown in turquoise color, those increasing in abundance are shown in red. Edge width is proportional to the change in abundance. Remaining lineages in the phylogeny of bacteria, archaea, eukarya, and some viruses are shown in light gray. PC1 shown at left, PC2 at right.

The nature of edge PCA lends itself to an intuitive inspection of the phylogenetic lineages explaining the difference in community structures. PhyloSift, by using pplacer’s guppy program and the Archaeopteryx tree viewer, can produce a visualization of the lineages most strongly associated with each principal component. Figure 3 shows this visualization for the edge PCA analysis of 106 fecal metagenome communities. In that figure, lineages are thickened proportionally to their contribution to the principal component, and are colored according to whether they increase (red) or decrease (turqoise) in abundance along the principal component axis. As we can see from Fig. 3 left, the first principal component is defined by an increase in Ruminococcacae, Clostridiales, and Bacteroides, with a decrease in Bifidobacteria. The association with age suggests that as communities develop in aging children, the Bifidobacteria become less abundant and members of those other lineages grow in abundance. The analysis of Yatsunenko et al. on 16S rRNA data also identified age-associated increases in Ruminococcacae and Bacteroides and a decrease in Bifidobacteria.

Table 1 Procrustes distances between microbial community analysis methods.

Analysis of 16S amplicon sequences with QIIME (QIIME 16S Amp) produces results more similar to PhyloSift analyzing either 16S or elite protein sequences from metagenomic data than to QIIME analysis of 16S sequences from metagenomic data. PhyloSift results for 16S and elite proteins are more similar to each other than to either QIIME method, possibly due to differences between Edge PCA and the QIIME-generated PCoA on UniFrac distances.

	QIIME 16S Meta	PhyloSift 16S Meta	PhyloSift Elite Meta	
QIIME 16S Amp	0.5134279	0.3873677	0.3762175	
QIIME 16S Meta	-	0.5376786	0.6351224	
PhyloSift 16S Meta	-	-	0.2450837	

Whereas the first principal component agrees strongly with the analysis reported by Yastunenko et al., the second principal component appears to identify a previously unreported aspect of variation in these samples. Extreme samples on the 2nd principal component (PC2) are very young infants whose fecal microbiota appear to be dominated not by Bifidobacteria, but instead by members of the genus Enterobacter and family Lactobacillales (see Fig. 3, right). One possible explanation for this observation may be an association with breast-feeding status of the infants. However, inspection of publicly available metadata did not reveal any clear association of PC2 with breastfeeding status or other recorded metadata. Another possible explanation is mode of birth, vaginal or caesarian, however no information on mode of birth is available for this dataset (J Gordon, pers. comm., 2013). We note that members of the Lactobacillales are abundant in the human vaginal tract, suggesting that newborns high on the 2nd principal component axis may be vaginally delivered if the two groups of newborns do indeed reflect differences in mode of delivery. Interestingly, the dimensions of community structure variation identified in the current set of 106 samples differ from those identified by Yatsunenko et al. in the larger set of 600 samples for which amplicon data are available. Geography and age were associated with most variation in their analysis of >600 samples, and the 106 metagenome samples are primarily from infants and do not equally represent that variation. It seems that age-related variation in the microbiome dominates the 106 metagenome samples.

We also investigated the diversity of microbes in the fecal samples. Classic measures of species diversity such as alpha and beta diversity have been applied to microbial communities by collapsing sequences to operational taxonomic units (OTUs). More recently, phylogenetic diversity (PD) (Faith, 1992) has been applied to metagenomic data, yielding a diversity metric that does not require defining OTUs (Kembel et al., 2011). In the present work we compute phylogenetic diversity on the placed reads, using the attachment points of reads to the reference tree as the basis for the diversity calculation. Figure 4 shows the phylogenetic diversity present in the fecal samples as a function of age. We observe a general trend where phylogenetic diversity grows quickly with age, presumably due to colonization of the infant gut, then continues to grow slowly throughout adult life. There is a significant log-linear relationship of phylogenetic diversity with age (Pearson’s product-moment correlation, p < 10−15). We also plot a variant of the PD metric called balance-weighted phylogenetic diversity (McCoy, Matsen & Frederick, 2013), where diversity contributed by each lineage is weighted by its relative abundance. Balance-weighted PD exhibits a similar growth in early life, but values for individual samples shift relative to population median values. Notably, balance-weighted PD declines in old age, suggesting that a smaller number of divergent lineages may come to dominate the adult human gut. The maximum balance-weighted PD value observed among any sample in the dataset was at the 7th month of life. When samples from before and after the 7th month of life are tested separately, balance-weighted PD exhibits significant age-associated growth before the 7th month (p = 0.009, Spearman’s rank correlation) and age-associated decline after the 7th month (p < 10−5, Spearman’s rank correlation). It is not clear what drives the reduction in balance-weighted PD after the 7th month of life, though we note that solid food is commonly introduced to the infant’s diet around this time.

Figure 4 Relationship between fecal community phylogenetic diversity and host age.

106 metagenomic samples were processed using PhyloSift and their phylogenetic diversity analyzed using two metrics. Unweighted phylogenetic diversity (PD) simply measures the total branch length of the reference tree covered by placed reads from a sample. Balance-weighted phylogenetic diversity adjusts these values by the abundance of each lineage in the sample. In unweighted PD, a log-linear relationship between host age and fecal community phylogenetic diversity can be observed. Balance weighted PD, on the other hand, shows rapid growth in early life followed by slow decline after the first year, consistent with a small number of divergent lineages becoming dominant in the fecal ecosystem.

PhyloSift provides a means to visualize the relative abundance of taxonomic groups present in a sample. Figure 5 shows two such plots for samples from a 1 month old breastfeeding infant and a 45 year old mother from the Yatsunenko et al. data (Yatsunenko et al., 2012).

Computational efficiency

When processing large metagenomic datasets, computational efficiency and resources can become a logistical challenge. For Illumina data, PhyloSift can process sequence reads on a single CPU core at least as quickly as they can be generated by current instruments. Figure 6 gives memory and running time requirements for some test Illumina datasets. The majority of PhyloSift’s running time is spent in phylogenetic read placement (data not shown). Most stages of the workflow implemented by PhyloSift are amenable to both fine and coarse-grain parallelism, thus parallel implementations of the workflow could be created should future data volumes demand it. Finally, the peak memory usage recorded during each run remains roughly constant at 6–9 GB across all data set sizes. As such, PhyloSift is memory-efficient enough to process metagenomic datasets on modern laptop hardware, wherein configurations with 8 GB RAM are readily available.

Figure 5 Taxonomic visualization of two human gut samples.

Taxonomic plot at left shows an infant, plot at right shows a 45 year old mother. Data analyzed by PhyloSift, visualized by Krona.

Figure 6 PhyloSift performance and scaling behavior.

PhyloSift v1.0 was used to process Illumina sequence data from a human gut microbiome dataset subsampled to varying numbers of reads. The program was run single-threaded on an Intel Xeon E5520 CPU core (circa 2009 model).

Discussion

We have presented a new approach for phylogenetic analysis of genomes and uncultured microbial communities. The software implementation of our method, called PhyloSift, also provides a platform for comparison of community structure among many samples. Phylogenetic analysis (placement of short sequences onto reference phylogenies) offers a number of conceptual advantages over OTU-based or taxonomic analysis (interpreting sequence data on the basis of hierarchal classification information) for metagenomic data. Without applying phylogenetic analysis, taxonomic analysis can produce results that are difficult to interpret, particularly when an unknown environmental sequence contains many high scoring hits to reference database sequences as is common in BLAST-based approaches. Alternatively, taxonomic information can be misleading for sequences from species lacking close relatives in public sequence databases; these sequences may recover no match at all, or be assigned taxonomic annotations which do not accurately reflect phylogenetic relationships (e.g., the closest match is still a distant relative, as reflected by low BLAST scores) (Eisen, 1998). Phylogenetic analysis avoids both of these problems, relying instead on evolutionary models to accurately place unknown sequences within a known topology. In many cases, phylogenies will also offer a higher resolution representation of genetic ancestry than taxonomies. For these reasons, we focus on types of phylogenetic analysis enabled by PhyloSift and forgo a discussion of previous taxonomy-based metagenome analysis methods.

Phylogenetic analysis of metagenome sequence data could in principle offer several advantages in the area of microbial forensics. First, by studying an uncultured community, some potential pitfalls of culture bias and sample contamination can be avoided entirely. Second, the environmental shotgun sequencing approach can avoid problems related to PCR primer bias, though issues related to DNA extraction bias remain a problem (Morgan, Darling & Eisen, 2010) and might be especially relevant for sporulating organisms such as the Bacilli and their relatives. Third, the metagenomic approach can be applied without prior knowledge of which genes to target in the sample, and permits interrogation of both slow-evolving genes such as 16S rRNA and fast evolving genes that might offer greater resolution among closely related organisms. Finally, phylogenetics can be applied to any gene of interest regardless of whether its evolutionary history is concordant with a taxonomic hierarchy.

Here we have introduced a means to statistically test for lineages of interest directly from an uncultured DNA sample. The test calculates a Bayes factor for the two competing hypotheses: zero sequences derive from the target lineage, versus one or more sequences in the sample derive from the target lineage. This method can be applied to any protein-coding or noncoding gene family of interest. Certain gene families will yield more sensitive tests than others, for example the 16S rRNA gene is slow-evolving and can not usually distinguish within-species relationships where some protein-coding genes might have greater resolution. We emphasize that the Bayes factor is not a test of homology – homology tests exist as e-value and related score statistics in aligners such as BLAST, LAST, and HMMER. Given sequences homologous to a gene family, the Bayes factor tests from which lineage they diverged. The limit of detection for this method will depend on how deeply a sample has been sequenced. This value will depend on several factors specific to the sequencing chemistry and currently must be calculated independently by the user.

The 37 elite gene families were selected because they are universally present and almost always in single copy, but there are some exceptions. When partial homologs exist interpretation of the lineage test can become complicated by paralogs or ancient lateral gene transfer events. Thus one must exercise appropriate caution when interpreting the results of the lineage test. It is a test of whether the sample is void of DNA predicted to have derived from a particular lineage in the phylogeny. For applications like medical diagnostics a more elaborate Bayesian hypothesis test might be appropriate. Such a test might check for a collection of genes that are diagnostic of the organism rather than seeking a single gene, based on prior knowledge that most of the 37 genes are present in most lineages. Such an approach would be less sensitive to sporadic lateral gene transfer events in any single gene family and represents a direction for future work.

Although we do not provide examples, it is possible to test the hypothesis that two microbial communities have equal composition using the phylogenetic Kantorovich-Rubenstein distance (Evans & Matsen, 2012). In a bioforensics context this approach could be applied to test whether two uncultured communities of interest “match” each other. The implementation of the method employs an efficient approximation to calculate p-values for the null hypothesis of equal community composition and has been described elsewhere (Evans & Matsen, 2012). This test can be applied directly to any individual gene family processed by PhyloSift or to the concatenated alignment of elite families at either the amino acid or DNA sequence level. One limitation of this test is that it does not currently provide a means to account for variability in apparent community structure introduced by normal sample handling procedures. Future work might develop tests that employ many technical replicates of samples to account for such variation in the hypothesis test.

PhyloSift can also be applied to explore the variation in community structure present in a collection metagenomic samples. In recent years it has become standard practice to explore microbial community structure variation using amplicon sequencing of highly conserved genes such as 16S rRNA, 18S rRNA and ITS regions followed by analysis with a pipeline such as QIIME (Caporaso et al., 2010), VAMPS (http://vamps.mbl.edu), or mothur (Schloss et al., 2009). Analysis of community structure using metagenome sequence has some potential advantages, such as avoiding issues related to PCR primer bias and distinguishing between erroneous PCR chimeras and sequences representing the “rare biosphere” (Bik et al., 2012). However, there are also shortcomings, such as the relatively sparse phylogenetic diversity of available reference genomes relative to amplicon databases. The reference-based approach taken by PhyloSift will suffer this database resolution limitation when processing metagenomic data, although not when processing amplicon data. Efforts to increase the phylogenetic diversity of available genome sequences are ongoing (Wu et al., 2009; Rinke et al., 2013; Shih et al., 2013)

Advances in the preparation of high throughput samples will make comparative metagenomics more tractable. The analysis we describe of human fecal microbial communities was possible with a median of only 50 Mbp sequence data per sample. Current Illumina HiSeq 2000 instruments generate up to 40 Gbp per lane, suggesting that up to 800 samples could be processed in a single Illumina lane and yield similar findings. Based on current Illumina sequencing service provider costs this suggests large-scale gut metagenome surveys could be conducted for as little as to $2.50 to $5 per sample in sequencing costs. Library preparation would dominate the overall cost of such studies, as current kits from Illumina require about $37 per sample.

Although we focus on phylogenetic analysis in this work, PhyloSift also provides a basic mechanism to attach taxonomic labels to branches of the phylogenetic trees. Our approach for taxonomic labeling of the phylogeny does not enforce a strict 1:1 mapping between taxonomic labels and branches in the phylogeny. Rather, each branch in the phylogeny is labeled with the entire set of most topologically consistent taxonomic labels. In cases where gene trees may be discordant with the taxonomic tree, this approach allows PhyloSift to represent some of this ambiguity in its results. A systematic study investigating the relationship between rates and patterns of LGT and the effectiveness of our approach for taxonomic labeling remains as future work, as does extension of the taxonomic labeling method to gene families for which duplication and loss is prevalent.

One major limitation of the current approach is that all phylogenetic analysis is conducted independently on each gene. However, genes do not evolve in isolation but rather co-evolve with each other in genomes. Recent studies have demonstrated that large parts of the phylogenetic history in different microbial genes are congruent even though they have undergone lateral gene transfer, duplication, and loss (Szãllåsi et al., 2012; Boussau et al., 2012). Large-scale statistical inference of phylogenetic networks (e.g., on >1000 microbial genomes) that account for duplication, transfer, and loss histories have not yet been described in the literature, however if such a network could be constructed it might provide a means to co-analyze all genes and yield a corresponding increase in sensitivity and power for statistical tests.

Availability

Software for Linux and Mac OS X, along with source code is freely available from http://github.com/gjospin/PhyloSift. Extensive user documentation is available at http://phylosift.wordpress.com. The source code has been licensed under the GNU Public License (GPL) v3.0.

Additional Information and Declarations

Competing Interests

Author Contributions

Data Deposition

Jonathan A. Eisen is an Academic Editor for PeerJ.

Aaron E. Darling conceived and designed the experiments, performed the experiments, analyzed the data, contributed reagents/materials/analysis tools, wrote the paper.

Guillaume Jospin performed the experiments, analyzed the data, contributed reagents/materials/analysis tools.

Eric Lowe helped with software testing.

Frederick A. Matsen IV performed the experiments, contributed reagents/materials/analysis tools, wrote the paper.

Holly M. Bik performed the experiments, analyzed the data, wrote the paper.

Jonathan A. Eisen conceived and designed the experiments, wrote the paper.

The following information was supplied regarding the deposition of related data:

FigShare, http://dx.doi.org/10.6084/m9.figshare.650869.

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
