# Peer review of "PhyloSift: phylogenetic analysis of genomes and metagenomes"

_PeerJ, doi:10.7717/peerj.243_

## Round 0.1 · original submission · Minor Revisions

It would be very helpful to include in the introduction an overview on other methods and tools for taxonomic assignment and metagenomic analysis. It would also be very interesting to conduct a benchmarking against other tools in terms of the accuracy of the taxonomic assignment and the time (and maybe space) complexity. Both reviewers commented on the use of conversational and colloquial wording; therefore, it is suggested to revise into a more formal phrasing. Minor issues: [1] 16S (not 16s) and [2] The specification of the CPU referred to in the manuscript might provide some perception of the runtime requirement of PhyloSift.

Reviewer 1 ·

Basic reporting

Writing is fairly conversational. For example: "every kind of natural and unnatural site" and "the diversity of kinds of microbes". While these examples are from the first paragraph, this is common through-out the paper. I recommend that the authors review the paper closely for style.

Lines 29-30: issues of contamination and taxonomic bias are still present in metagenomic surveys. For example, human skin is a common contaminant of microbial samples, and any given DNA extraction procedure will work better for some taxa than others. The real benefits of metagenomics are reduced cost of sequencing in terms of time, labor and money. I recommend revising this section for accuracy.

The term "random shotgun metagenomic data" is introduced on line 59, but isn't defined there. It is worth defining this as the approaches described earlier for readers who aren't as familiar with this field.

Line 136: "During database construction (described elsewhere)" - mention where this is described. Is this database publicly accessible as well? It would be useful for researchers interested in comparing other methods against PhyloSift if the reference data is available independently of the software.

Line 144 is confusing. I suggest rewording.

Line 156: discuss the shortcomings.

Line 158: I'm confused about how this happens. How do you know which specific aligned reads of elite gene A match up with (i.e., are from the same genome as) which specific reads of elite gene B?

The authors should provide the run times of their methods on different sized data sets so readers can get an idea of how long they will take to run, and how they scale.

Experimental design

No comments.

Validity of the findings

Figure 2: Is the QIIME analysis based on PCA or PCoA? If PCA, it should be recomputed as PCoA of unifrac distances, but I think it's PCoA based on line 329. The plots look similar, but Procrustes analysis would allow the authors to quantify the similarity (see for example this paper: http://www.sciencemag.org/content/332/6032/970.abstract).

Figure 4: A regression analysis is in order here. Also, while the results are different, it's not clear which is better.

The Greengenes database is becoming the most widely used and reliable source for bacterial taxonomy. In addition to the NCBI taxonomy, it would be extremely useful if PhyloSift could provide taxonomic summaries based on the Greengenes taxonomy.

When concatenating alignments of elite gene family reads, how is missing data handled (ie., families that aren't read)? If these are added as all gaps, how does that effect the quality of the resulting tree? (I'm not challenging the method here - I'm unsure of how it would work.)

Line 358: The authors say that a possible explanation is breast-feeding status, but in the next sentence say that that does not appear to be the case. I'm confused about this. The authors next speculate about delivery mode, and note that there is no metadata available to investigate that. I would recommend getting in touch with the authors of the original paper to see if that data is available.

Additional comments

In general I think this is a useful addition to the field. As a general comment, I think the software is more likely to be well-received if the authors demonstrated novel results that could be identified from existing data using these methods which could not be obtained otherwise. Currently the authors apply their methods to the Yatsuneko Nature paper data. They highlight a different result for alpha diversity with their weighted PD metric, but that metric is not the focus of the paper (it sounds like this is reported somewhere else as the method is cited as in review). For the difference they observe in beta diversity with respect to QIIME, it's not clear whether this difference is something of biological interest, or an artefact of their methods as they're not able to provide any explantation for why it occurs.

·

Basic reporting

The article appears to adhere to the PeerJ policies. The introduction is currently rather under-referenced, there is only a single reference on the first page! I would appreciate it if the authors would provide the necessary references for key concepts in the introduction; relevant metagenomics review articles, examples of case studies in the various fields mentioned, etc. Given the manuscript describes a pipeline for metagenomic sequence analysis, I would like to see more space devoted in the introduction to outlining the major methods for analysing such data (similarity-based, phylogenetic and composition based methods) and software implementations of these methods. This should perhaps be prioritised over some of the earlier introduction which could be edited for concision. The manuscript sometimes slips into colloquial use of language, e.g. 'more vexing'.

Minor items


Figure 1 is a little confusing in that it looks like the labels "rRNA" and "protein" relate to the input, whereas my impression is that the input is all metagenomics DNA which is translated.


I think it is not strictly true to say "First, metagenomic samples reflect entire communities of organisms, unlike “traditional” genome sequencing which reflects a single individual of a population." as whole-genome shotgun sequencing often sequences a population that may or may not be clonal, depending on the way the sample was prepared (e.g. single cell, single colony, multiple colonies, multiple cells) etc.

"Loss of linkage information occur in two ways: during sample extraction and size selection of fragments for sequencing." would be more accurate to say during fragmentation, as size selection is an optional step.

In Figure 2 the PCA plots could use a legend delineating the meaning of the colours.

Experimental design

This is my first review for PeerJ and I am not sure how the guidelines should be interpreted for journal articles describing software. However, given the paper is structured into the standard Introduction/Methods/Results/Discussion format, I think the manuscript would benefit from some restructuring to ensure that methods and results are correctly placed. It may be helpful, as per the editorial guidelines, to explicitly state a question or set of questions which are addressed in the manuscript. For example, one approach might be to compare to existing methods, with the aim of providing a rationale for why Phylosift is superior in some regards to other solutions, either in terms of accuracy of phylogenetic placement, running time, additional functionality, etc. A table with some comparisons to other commonly used software would be really helpful. Personally I am interested in comparisons with approaches such as MEGAN (with LCA), Metaphlan, Phylopithia, ribosomal MLST and simpler approaches such as BLAST-best-hit.

Validity of the findings

I downloaded the software and successfully ran it both on the Mac and on the PC, with a few minor problems (for example, the shell script to launch Phylosift didn't work on the Mac, and I had to call it from the bin/) directory. I liked the selection of default reports including the taxonomic breakdowns and the Krona images which were intuitive to use. The results on my yet samples were promising, although I found that species and subspecies assignments were not always accurate. I appreciated the statistical information which i could use to interpret the confidence in these results. The software was rather slow to run on 1000 and 10,000 reads. I wonder how well this approach will scale. I appreciate the suggestion of the use of contigs to speed the process up, although I worry about the loss of abundance information, and I also worry about the formation of chimeric, 'consensus' contigs from mixtures that obscure rather than improve phylogenetic signal. I did find a very erroneous assignment of a read from Bacteroides hitting instead Vibrio cholera with strong statistical report which I have reported to the authors who are looking into it. On the basis of my findings I would have really liked to have seen a comparison of taxon assignment accuracy compared to other pipelines, perhaps trilled on mock community data. I was impressed by the level of documentation, the Github repository for the software and evidence of strong support from the developers.

Additional comments

My major recommendation is that manuscript is improved for readability, by restructuring by reference to specific question or questions, and ensuring the correct separation of methods and results.

---

## Round 0.2 · accepted · Accept

Thank you for the revised version of the manuscript. The manuscript is suitable for the publication in PeerJ.

·

Basic reporting

I am grateful for the authors' considered responses to my questions and comments and am happy with the revised manuscript.

Experimental design

No comments

Validity of the findings

No comments